# Design of a 3D Amino-Functionalized Rice Husk Ash Nano-Silica/Chitosan/Alginate Composite as Support for Laccase Immobilization

**DOI:** 10.3390/polym15143127

**Published:** 2023-07-22

**Authors:** Francesca Romana Scuto, Clarissa Ciarlantini, Viviana Chiappini, Loris Pietrelli, Antonella Piozzi, Anna M. Girelli

**Affiliations:** 1Department of Chemistry, Sapienza University of Rome, P. le A. Moro 5, 00185 Rome, Italy; francescaromana.scuto@uniroma1.it (F.R.S.); clarissa.ciarlantini@uniroma1.it (C.C.); viviana.chiappini@uniroma1.it (V.C.); antonella.piozzi@uniroma1.it (A.P.); 2DAFNE Department, Tuscia University, Via Santa Maria in Gradi 4, 01100 Viterbo, Italy

**Keywords:** rice-husk ash, nano-silica, immobilized laccase, composite, APTES, syringic acid removal

## Abstract

Recycling of agro-industrial waste is one of the major issues addressed in recent years aimed at obtaining products with high added value as a future alternative to traditional ones in the per-spective of a bio-based and circular economy. One of the most produced wastes is rice husk and it is particularly interesting because it is very rich in silica, a material with a high intrinsic value. In the present study, a method to extract silica from rice husk ash (RHA) and to use it as a carrier for the immobilization of laccase from Trametes versicolor was developed. The obtained mesoporous nano-silica was characterized by X-ray diffraction (XRD), ATR-FTIR spectroscopy, Scanning Elec-tron Microscopy (SEM), and Energy Dispersive X-ray spectroscopy (EDS). A nano-silica purity of about 100% was found. Nano-silica was then introduced in a cross-linked chitosan/alginate scaffold to make it more easily recoverable after reuse. To favor laccase immobilization into the composite scaffold, functionalization of the nano-silica with (γ-aminopropyl) triethoxysilane (APTES) was performed. The APTES/RHA nano-silica/chitosan/alginate (ARCA) composite al-lowed to obtain under mild conditions (pH 7, room temperature, 1.5 h reaction time) a robust and easily reusable solid biocatalyst with 3.8 U/g of immobilized enzyme which maintained 50% of its activity after six reuses. The biocatalytic system, tested for syringic acid bioremediation, was able to totally oxidize the contaminant in 24 h.

## 1. Introduction

Enzymatic immobilization is currently a technique of great novelty and interest in the scientific community since it can highly increase the economic convenience of a biotechnological process by improving the pH and temperature stability of enzymes and permitting biocatalyst recovery and reuse [1]. Several enzymes find extensive use in enzyme immobilization, depending on the specific application, but the most commonly employed include lipases and laccases [2]. To date, laccases have shown high potential in the degradation of several organic pollutants [3,4,5,6,7,8,9], and their use is considered an eco-friendly approach to protect the environment and an alternative method to existing technologies. For this reason, the possibility of improving the performance of laccases through immobilization is an extremely appealing area, and its investigation continues to be crucial for novel biotechnological developments and industrial applications.

In recent years, many efforts have been made to diversify immobilization methods, especially by developing new types of inexpensive and eco-friendly support that have the characteristics necessary for good immobilization (high porosity and affinity to enzymes, presence of reactive functionality, recyclability, thermal stability or possibility to work in organic solvents, etc.) [10]. For this purpose, research has focused on the use of biocompatible polymers, such as collagen, gelatin, chitin, hyaluronic acid, cellulose, dextran, glycosaminoglycans, and many others [11], thanks to their easily modulable chemical-physical properties and the possibility of modeling them in different shapes able to allow a high interaction between the biocatalyst and the support (particles, membranes, nano-fibers, hydrogels, scaffolds). Among these natural polymers, chitosan has gained significant attention in the last two decades [12,13]. This interest derives from its ease of processing, which allows obtaining porous structures simply and economically, and its biocompatibility, biodegradability, and low immunogenicity. Most of these peculiar properties derive from the presence of amino groups along the chitosan backbone, which also makes it suitable for being derivatized for any application [14]. However, being a polysaccharide, chitosan needs to be cross-linked with a plethora of different molecules [15,16,17,18] to allow dimensional stability to be achieved in an aqueous environment, which is fundamental in the reusing process of the solid biocatalyst. Among them alginate, a linear unbranched polymer containing β-(1→4)-linked D-mannuronic acid and α-(1→4)-linked L-guluronic acid [19], is one of the most used [20,21]. Anyway, even when cross-linked, natural polymers can be prone to swelling and degradation, especially under harsh operating conditions, which may lead to the loss of enzyme activity and reduced lifespan of the immobilized enzymes. For this reason, an in-depth study of this type of support is still necessary, and careful characterization, modification, and optimization are necessary to guarantee good stability and a high surface area to ensure their successful application in enzyme immobilization.

One method to improve the features of natural polymeric supports can be the addition of a filler such as silicon carbide, aluminum oxide, zinc oxide, graphite, zeolite, calcium carbonate, and many others [22]. For enzyme immobilization, nano-silica seems to be the ideal filler as it presents the advantages of having small pores, large specific surface area, strong surface adsorption, and good dispersion [23]. However, the traditional silica extraction methods are reported to be hazardous to the environment as they involve the intensive use of chemicals, energy, and nonrenewable resources [24]. For this reason, attempts have recently been made to find alternative feedstocks for its production [25]. In this context, the exploitation of a lignocellulosic waste such as rice husk (RH) for nano-silica synthesis holds significant importance because it brings many advantages, such as the reduction in pollution and cost management related to the waste disposal and the minimization of the environmental impact of silica extraction. RH disposal management is a serious problem that needs to be addressed as about 150 million tons of this waste are generated annually [26]. Furthermore, its uncontrolled incineration and disposal in open fields or landfills may create both environmental and human health problems due to the low density of the ashes (2.22 g/cm^3^) [27]. In addition, the use of RH for silica extraction is a largely favorable process, as among the various lignocellulosic wastes, it is certainly among those with the highest percentage of ash [28]. RH ash (RHA) is the byproduct of RH incineration and is particularly rich in silica (up to 95%) [29]. From this waste is possible obtaining three types of silica depending on the RH incineration temperature: amorphous (T < 600 °C), crystalline (T > 800 °C), and semi-crystalline silica [30]. For this reason, RH is a suitable candidate as feedstock of a certain industrial interest also considering the advantage of its low cost. RHA is currently used in industries associated with ceramics, electronics, catalysis, pharmaceutics, and other materials [31,32], but considering a circular economy approach, the recovery and reuse of RHA requires continuous and in-depth investigation of novel potential applications such as enzyme immobilization.

To sum up, the main aim of this study was to synthesize a robust, cheap, and biocompatible support suitable for the immobilization of laccase from *Trametes versicolor* based on a natural polymeric composite reinforced with nano-silica derived from a recycled source. In particular, a chitosan/alginate polyelectrolyte complex was cross-linked with CaCl_2_ to obtain a macro-porous three-dimensional (3D) solid scaffold. RHA-derived nano-silica was then introduced in the scaffold to improve its thermal and mechanical properties and functionalized with (γ-aminopropyl) triethoxysilane (APTES). The functionalization with APTES was aimed at improving the immobilization efficiency by increasing ionic and hydrogen interactions between the enzyme and support through the introduced amino groups. The use of a 3D structure based on silica-reinforced polymers instead was aimed at guaranteeing easier handling of the biocatalyst, larger active surfaces necessary for a greater enzymatic load, as well as better stability of the solid biocatalyst under varying experimental conditions. Finally, to verify its possible use for wastewater treatment, the solid biocatalyst was tested in the degradation of syringic acid, a chemical marker of biomass combustion.

## 2. Materials and Methods

### 2.1. Materials

The rice husk used in this study was provided by Riseria Roncaia Romano snc, Mantova, Italy. Commercial silica (Nucleosil, 100 Å-5 micron) was purchased from Macherey-Nagel (Düren, Germany). Laccase from *Trametes versicolor* with a nominal activity of 136 U/mg protein, 2,2-azinobis(3-ethylbenzothiazoline-6-sulfonicacid) diammonium salt (ABTS), chitosan (medium molecular weight, 75–85% deacetylated), sodium alginate (viscosity 5–40 cps) and salts for buffer solutions were purchased from Sigma-Aldrich (Milan, Italy). All the other chemicals including hydrochloric acid (HCl, 37%), NaOH, Na_2_CO_3_, (γ-aminopropyl) triethoxysilane (APTES) were provided by Merck Life Science Private Ltd. (Darmstadt, Germany).

### 2.2. Purification and Calcination of Rice Husk

Generally, RHA nano-silica includes several mineral impurities such as K, P, Fe, Mn, Mg, Cu, Zn, Na, Al, Cl, and S. Therefore, it is necessary to remove these compounds to obtain pure nano-silica [33,34]. RH was washed several times with distilled water to remove possible impurities and dried at 60 °C for 3 h. Then, 10 g of dried RH was treated with 1 M HCl at 75 °C for 1 h to remove metals, minerals, and pigments. Afterward, it was washed with deionized water and dried at 60 °C for 3 h. Finally, the purified rice husk was pyrolyzed in a muffle furnace at 550 °C for 5 h using a ramp rate of 15 °C/min to obtain rice-husk ash (RHA).

### 2.3. Silica Extraction from Rice Husk Ash

Nano-silica extraction from RHA was carried out following the procedure reported by Nayak and Datta [34]. Briefly, 5 g of RHA was treated with 30 mL of 0.5 M Na_2_CO_3_ in a water bath at 90 °C for 2 h under continuous stirring. Then, the sample was cooled and neutralized with 1 M HCl (pH value was monitored with a Crison pHmeter) and left to settle for 24 h at 4 °C. The chemical reaction of the nano-silica extraction is reported in Figure 1.

The sample was then washed with a suitable amount of water and centrifuged several times at 5000 rpm for 15 min to recover the nano-silica extracted. The nano-silica was finally dried at 80 °C for 3 h. The whole recovery procedure was carried out in triplicate to calculate, using Equation (1), the recovery percentage of nano-silica from RHA.
(1)$$\mathrm{Nano}\text{-}\mathrm{silica}\;\mathrm{recovery}\;(\%)=\frac{\mathrm{mass}\;\mathrm{of}\;\mathrm{the}\;\mathrm{obtained}\;\mathrm{silica}}{\mathrm{RHA}\;\mathrm{initial}\;\mathrm{mass}}\times 100$$

### 2.4. Chitosan/Alginate (CA) Scaffold Preparation

The scaffolds were synthesized by forming an electrolytic complex with the two natural polymers, chitosan and alginate, cross-linked with CaCl_2_ to ensure the stability of the structure in an aqueous medium and to improve the mechanical properties of the 3D scaffold. To prepare the chitosan/alginate (CA) scaffold, 0.6 g of chitosan and 1.2 g of sodium alginate were dissolved in 20 mL of 1 M acetic acid (2% *v*/*v*) and 30 mL of 1 M NaOH, respectively. The two solutions were mixed and maintained under continuous stirring for 24 h to obtain an electrolytic complex of 3% chitosan and 4% alginate. The pH of the chitosan/alginate solution was adjusted to 7.4 by adding 2 M acetic acid dropwise, avoiding the dialysis step. The mixture was then frozen for 24 h and lyophilized. Afterward, the tridimensional system (scaffold) was cross-linked with a 3% (w:v) CaCl_2_ solution for 1 h. Subsequently, the scaffold was washed several times and then immersed for 20 min in deionized water to remove the unbound CaCl_2_. Finally, the sample was dried in a freeze-dryer.

### 2.5. RHA Nano-Silica/Chitosan/Alginate (RCA) Scaffold Preparation

An interesting approach for the preparation of new materials is to combine polymers with inorganic components to synthesize composite scaffolds. Following this current, RHA nano-silica was introduced into the scaffold by imbibition method: 5 mL of silica suspension (18 mg/mL) was inserted into a tube containing the CA scaffold (13 mm diameter × 12 mm height) to completely cover the scaffold and to obtain an RHA nano-silica/scaffold ratio of 2:1 (w:w). The system was kept for 1 h at room temperature under gentle stirring to ensure the silica diffusion into the pores of the scaffold. The scaffold was then rinsed several times with a water/ethanol solution for 20 min to remove unbound silica. Finally, the composite scaffold (RCA) was frozen and then lyophilized for 24 h.

The RHA nano-silica presence in the RCA scaffold was verified by comparing the intensities of the IR bands around 1050 (Si–O–Si stretching) and 800 cm^−1^ (Si–O stretching) of the imbibed scaffolds (1:1 and 2:1 nano-silica:scaffold ratios (w:w)) with those of the CA scaffold.

### 2.6. Functionalization of RHA Nano-Silica/Chitosan/Alginate (RCA) Scaffold with APTES

The APTES is a reagent generally used to promote adhesion between silica-based substrates and organic or metallic compounds in the development of advanced materials [35,36,37]. To determine the optimal APTES amount needed for the subsequent enzyme immobilization reaction, preliminary experiments were carried out on the filler alone (RHA nano-silica) before its introduction into the scaffold. Different APTES:RHA nano-silica ratios (1:2; 1:1; 2:1; 5:1, 10:1 (w:w)) were examined. The protocol is described as follows: 40 mg of RHA nano-silica was added to 1 mL of ethanol containing from 0.084 to 0.42 mL of APTES to obtain the desired APTES:RHA nano-silica ratio. The reaction was carried out under stirring for 3 h at room temperature. The resultant product was firstly washed and centrifuged for 10 min at 4000 rpm with water and then ethanol to eliminate the unbound APTES. Finally, the aminated RHA nano-silica (APTES/RHA nano-silica) was dried at 70 °C for 3 h and then stored in a desiccator with silica as a dried agent. ATR-FTIR and SEM-EDS analysis showed that the optimal ratio APTES:RHA-nano-silica was 5:1 (w:w).

Once the preliminary study was completed and the optimal ratio was determined, the scaffold was prepared with the same procedure described in Section 2.5. However, in this case, the amination step with APTES was performed after the RHA nano-silica imbibition. Finally, the APTES/RHA nano-silica/chitosan/alginate (ARCA) scaffold was frozen and then lyophilized for 24 h.

### 2.7. Samples Characterization

#### 2.7.1. Elemental and ATR-FTIR Spectroscopy Analysis

The elemental composition of the materials (C%, H%, N%) was determined using the EA 1110 CHNS-O analyzer (CE Instruments, Milano, Italy). Furthermore, they were also analyzed by Fourier-transform infrared spectroscopy (FTIR) before and after every synthesis step to identify the functional groups present on the surface. The spectra were obtained using attenuated total reflection (ATR) by a Nicolet 6700 (Thermo Fisher Scientific, Waltham, MA, USA) equipped with a Golden Gate single-reflection diamond ATR accessory at a resolution of 2 cm^−1^ and co-adding 200 scans.

#### 2.7.2. Scanning Electron Microscopy (SEM) Analysis

To characterize the morphological structure of untreated RHA, RHA nano-silica, APTES/nano-silica, as well as that of the scaffolds with or without imbibed RHA nano-silica, a scanning electron microscope (SEM; LEO 1450 VP; Carl Zeiss, Oberkochen, Germany) was used. All the samples were dried and chromium coated to make their surfaces conductive before the analysis. The micrographs obtained were taken at different magnifications (1–50 KX), 14 mm as working distance, 15 kV as accelerating voltage, and DISS as digital image recording.

#### 2.7.3. Powder X-ray Diffraction (XRD), BET Analysis, and Porosity Determination

The RHA nano-silica obtained was characterized by powder X-ray diffraction using a D8 Advance Diffractometer (Bruker, Karlsruhe, Germany) with the molybdenum Kα1 radiation (λ = 0.7071 Å). The XRD patterns were measured at room temperature with a step size of 0.01° in the small-angle range.

The surface area of RHA nano-silica, determined by the Brunauer–Emmett–Teller (BET) multipoint method and textural analysis, was obtained by N_2_ adsorption/desorption measurements at liquid nitrogen temperature (77 K), using a 3-Flex analyzer (Micromeritics, Norcross, GA, USA). Before analysis, the samples were pre-treated under vacuum at 200 °C for 2 h. The pore distribution was determined from the adsorption curve by the Barret–Joyner–Halenda (BJH) method [38] and from the analysis of the mesopore isotherm by the *t*-test considering the curve of Harkins and Jura. The total pore volume was determined by the Gurvitsch rule.

As a BET analysis of the whole scaffolds could not be obtained without causing damage to their structure; to have information about the porosity of the scaffolds, the liquid displacement method was used [39,40]. The scaffold of interest, with a certain initial weight (W_0_) and volume (V_0_), was dipped in 10 mL of ethanol for 30 min (ρ_EtOH_ = 0.789 g/cm^3^ at 20 °C). Ethanol was chosen as it is a non-solvent of chitosan and therefore does not involve swelling of the polysaccharide. After 30 min, the scaffold was removed from the solvent and weighed (W_1_). The porosity (p%) was calculated through the following Equation (2):(2)$$p\%=\left(\frac{W_{1}-W_{0}}{\rho_{EtOH}\times V_{0}}\right)$$

#### 2.7.4. Mechanical Testing and Thermal Stability of the Scaffolds

The compressive mechanical strength and modulus of cylindrical scaffolds with or without imbibed RHA nano-silica (13 mm diameter × 12 mm height) were determined by using an Instron 4505 mechanical tester (Instron Corp., Canton, MA, USA) with 2 kN load cells following the guidelines described in the ASTM D5024-95a method. The crosshead speed of the Instron tester was set at 0.4 mm/min, and the scaffold was compressed to approximately 30% of its original thickness.

Perkin Elmer TGS-2 Analyzer (PerkinElmer, Waltham, MA, USA) was used to obtain the thermal decomposition profile of the scaffolds. Approximately 2 mg of the sample was heated until 800 °C inside a platinum pan connected to an electrical microbalance. The heating was performed with a 10 °C/min ramp in O_2_ atmosphere and the weight loss was measured as a function of the temperature.

### 2.8. Laccase Immobilization on APTES/RHA Nano-Silica/Chitosan/Alginate (ARCA) Scaffold

To optimize the immobilization reaction conditions, laccase solutions with different free activity (U) were added to various amounts of ARCA scaffolds to obtain 0.17, 0.44, 0.75, and 0.77 U/mg ratios. A laccase solution was prepared in 1 mL 0.05 M phosphate buffer (PBS) at pH 7 making sure that the concentration was enough to have 0.77 U of laccase per mg of scaffold. Then the system was left to react for 1.5 h at 25 °C to allow the physical immobilization. After, the scaffold was removed from the reaction medium and washed several times with 0.05 M PBS at pH 7.

### 2.9. Free and Immobilized Activity Determination

The activity of the immobilized enzyme was evaluated by spectrophotometric assay employing ABTS as substrate. The analysis was performed by adding 2 mL of ABTS (0.18 mM) and 0.7 mL of a 0.1 M citrate/0.2 M PBS at pH 3 solution to 10 mg of the solid biocatalyst (V_tot_ = 2.7 mL). The reaction was followed by monitoring every 30 s for 5 min the oxidation of ABTS to its radical cation ABTS (molar extinction coefficient, ε = 36,000 L mol^−1^ cm^−1^) at 420 nm employing a Model T60 UV-Vis spectrophotometer (PG Instrument Limited, Leicester, UK). The same procedure was employed to determine the free enzyme activity using 10 μL of laccase solution instead of 10 mg of the biocatalyst. One activity unit was defined as the amount of enzyme needed to oxidize 1 μmol of ABTS per minute at 30 °C and pH 3 (0.1 M citrate/0.2 M PBS). The biocatalyst activity was calculated using the following Equations:(3)$$\mathrm{Free}\;\mathrm{laccase}\;\mathrm{activity}\;(U/\mathrm{mL})=\frac{∆A_{min}\;V_{tot}}{\varepsilon \;V_{w}}\;10^{6}$$
(4)$$\mathrm{Immobilized}\;\mathrm{activity}\;(U/g)=\frac{∆A_{min}\;V_{tot}}{\varepsilon \;m}\;10^{6}$$
where ∆A_min_ is obtained following the ABTS formation; V_tot_ is the total reaction volume during the activity determination (2.7 mL); V_w_ is the volume withdrawn from the sample (10 μL); m is the weight of the carrier (10 mg); and 10^6^ is a dimensional factor.

### 2.10. Evaluation of the Immobilized Biocatalysts

As reported by Sheldon et al. and Boudrant et al. [41,42], two parameters were employed to evaluate the immobilization procedure success: immobilized activity (U/g) and immobilization efficiency (%). The first parameter was calculated as described in Equation (4), while for determination of the other was used the following Equation:(5)$$\mathrm{Efficiency}\;(E\%)=\frac{U_{i}}{U_{0}-U_{f}}\times 100$$
where U_0_ and U_f_ are, respectively, the initial and the residual free enzyme activity utilized in the reaction mixture, and U_i_ is the immobilized enzyme activity on the carrier. All the above terms were calculated using total activity units (i.e., μmol/min) and not using specific activities (i.e., U/L, U/g).

### 2.11. Operational Stability and Syringic Acid Oxidation

The operational stability (or reuse) of the solid biocatalyst was evaluated by performing various cycles of ABTS oxidation as described in Section 2.8, always using the same biocatalyst and a total volume of 2.7 mL. After each cycle, the reaction solution was removed with a pipette and the biocatalyst was washed with 0.05 M PBS at pH 7. The relative activity of the biocatalyst was calculated for each cycle, assuming a 100% relative activity for the first cycle.

The phenol bio-removal study was carried out by adding 2 mL of a syringic acid standard solution (50 mg L^−1^) in 0.1 M citrate/0.2 M PBS (pH 5) to the solid biocatalyst (total immobilized activity 0.013 U). The syringic acid oxidation was monitored for 24 h both by spectrophotometric measurements (250–450 nm range) and HPLC-DAD system (Shimadzu, Milan, Italy). The HPLC system included a C18 column (15 cm × 4.6 mm) and H_2_O:MeOH (70:30 v:v) with a flow rate of 1 mL/min as mobile phase.

## 3. Results and Discussion

### 3.1. Characterization of RHA Nano-Silica

Since the RHA calcination time can influence the purity of the extracted nano-silica, this parameter was investigated. The optimal conditions were chosen considering the lowest values of N%, C%, and S% detected by elemental analysis measurements of the nano-silica samples calcined at different times: the lower these values, the higher the loss of impurities. The better purity of the nano-silica is also underlined by the increase in the whiteness of the material when lower values of C, H, and S are reached. The results, reported in Table 1, highlight that 5 h at 550 °C was a sufficient time to degrade almost all the organic matter.

To determine the purity of the extracted nano-silica, EDS analysis was carried out (Figure 2a). The presence of silicon (29.59 atomic %) and oxygen (69.55 atomic %) in the mole ratio of about 1:2, the absence of sodium and potassium or other elements, and the low carbon percentage (<0.1 atomic %) confirmed the purity of nano-silica.

These results are in accordance with those of Nayak et al. [26], who reported that when using Na_2_CO_3_ in the alkaline treatment, CO_2_ is generated during the neutralization step. This gas hinders the formation of a gel network structure that can entrap impurities, preventing their removal during the washing phase. Additionally, the EDS analysis demonstrated that 64.96% of the sample weight was attributed to silicon. This result was higher than the one reported by Dashan and Khadar [43] (48.36 weight % at 400 °C), indicating that 550 °C was an optimal temperature value as it permitted to obtain a large amount of pure nano-silica.

The RHA nano-silica structure was investigated by XRD analysis employing molybdenum as the target material. The absence of any sharp peaks, commonly related to a crystalline structure, indicated the amorphous nature of the RHA nano-silica synthesized (Figure 2b). The XRD patterns obtained in this study were similar to those of Bakar et al. [44] and Rafiee et al. [45], who also produced amorphous nano-silica. In particular, it was observed a broad diffused peak centered at the 2θ value of 10.1°, corresponding to that of 2θ = 22.3° found with copper as the target material, confirming the completely amorphous nature of the obtained nano-silica as the crystallization of acid-leached silica occurred above 900 °C [44].

The morphological structure of the various materials was investigated by SEM, as shown in Figure 3, and BET analysis. Before calcination, RH showed a well-organized, corrugated, and non-porous layered structure with silica localized between the folds of the layers (Figure 3a,b). On the contrary, RHA nano-silica was characterized by a highly porous structure (Figure 3c,d) and found to be mesoporous by BET analysis, with an average pore diameter of about 3.4 nm total pore volume of 0.225 cm^3^/g and surface area of 111 m^2^/g. The particles had a size of 70–100 nm, in agreement with the dimensions reported in the literature [45,46]. As reported by Fernandes et al. [47], different shapes of the particles and the presence of some clearly defined aggregates of layered flakes (Figure 3d) can be observed from SEM images. This last peculiarity was probably due to the considerable exposition of the hydroxyl (–OH) groups.

### 3.2. Characterization of RHA Nano-Silica/Chitosan/Alginate (RCA) Scaffold

Alginate was used for the preparation of chitosan-based composite scaffolds. This type of matrix was selected as it offers many benefits such as simple and cost-effective preparation procedures, good robustness, high immobilization capacity, biocompatibility, and possibility of employment in various biotechnological and biomedical applications [48,49]. The scaffolds were successively imbibed with RHA nano-silica, which was used both as mechanical reinforcement and as a binding surface for laccase immobilization.

In Figure 4A, the ATR-FTIR spectra of the scaffolds containing RHA nano-silica at two nano-silica:scaffold ratios, 1:1 and 2:1 (w:w), in comparison with those of CA and pristine nano-silica were reported. In the spectrum of the CA scaffold can be highlighted an intense band in the range 3600–3000 cm^−1^ attributed to the stretching of the –OH groups of the two polysaccharides, and two peaks at 1604 and 1545 cm^−1^ assigned to the carboxylic groups of the alginate which interact with chitosan (–COO^−^ antisymmetric stretching and antisymmetric bending of the protonated amine and amide I, at 1604 cm^−1^) and to the amino groups of chitosan which interact with alginate (N-H amine bending, amide II and symmetric –NH_3_^+^ bending, at 1545 cm^−1^). The carboxylic groups of the alginate also showed a symmetric stretching at 1410 cm^−1^. Finally, in the range 1180–800 cm^−1^, the C–O–C, C–O, and C–O–H stretching bands due to the pyranose rings were evidenced. After introduction of RHA nano-silica, some changes were observed in the spectra of the modified scaffold. First of all, the peaks attributed to the antisymmetric and symmetric stretching of the Si–O–Si group were evident at 1040 and 800 cm^−1^, respectively. Furthermore, the remarkable intensity decrease in the bands at 1410 cm^−1^ and 1545 cm^−1^, with the peak at 1545 cm^−1^ becoming a shoulder of the band at 1604 cm^−1^ (see magnification shown in Figure 4B), confirmed the hydrogen bonding interactions between the SiOH groups and carbonyl and amino groups of the CA scaffold. In particular, the decrease in the band intensity at 1545 cm^−1^ results correlated to the nano-silica content in the structure (scaffold with a nano-silica:scaffold ratio of 2:1) demonstrating an increase in the number of hydrogen bonds established between the SiOH and amino groups of chitosan with an increase in the filler.

By comparing TGA thermograms, it was possible to observe that RHA nano-silica is able to make the scaffolds more thermally stable (Figure 5), probably due to interactions between its –OH groups and the ones of the polymers (–OH, –NH, and –CO). The thermal degradation of every scaffold occurred in three stages: (i) dehydration of the polymers at about 100 °C, (ii) degradation of non-cross-linked alginate and chitosan chains, and (iii) degradation of the Ca^2+^-cross-linked chains. It is interesting to note that in the second stage, the degradation temperatures of the polyelectrolytic complexes were lower (220 °C) than the ones reported by Zhao et al. [50] (270 °C). This was probably due to the presence of Ca^2+^ ions, which interacting with –COOH groups of alginate, decreased the thermal stability. However, it was noted a significant increase in the total thermal stability, especially for the scaffolds imbibed with RHA nano-silica. Considering the shape of the TGA curves, at 280 °C the scaffolds reached approximately 77.8% of their weight, and for higher temperatures, the behavior started to diverge: at 580 °C the CA scaffold showed a notable loss in weight (about 67%), while better thermal stability was shown for both RCA scaffolds containing the filler at a 2:1 (w:w) and 1:1 (w:w) nano-silica:scaffold ratio (47.7% and 58% of weight loss, respectively). Moreover, the liquid displacement method measurements revealed only a slight porosity decrease (from 71% to 67%) when RHA nano-silica was imbibed to the system.

Finally, to verify the reinforcement effect of RHA nano-silica on the scaffolds, the compression modules were measured: 194 MPa and 241 MPa for the CA and RCA scaffolds were obtained, respectively. Therefore, it can be concluded that RHA nano-silica positively affected the mechanical and thermal properties of the scaffolds, without affecting their porosity.

### 3.3. Functionalization of RHA Nano-Silica/Chitosan/Alginate Scaffold with APTES

In this work, APTES was used to increase the amine group amount on the composite scaffold to favor the physical binding of the enzyme [51,52,53,54,55]. The RHA nano-silica amination was confirmed by ATR-FTIR spectra, where it was possible to observe an increase in intensity of the bands at 2919 cm^−1^ and 2800 cm^−1^ and the formation of a new peak at 1310 cm^−1^ (due to contribution of CH_2_ stretching and bending of propyl chain in APTES, respectively) [50]. To estimate the amination yield, the intensity ratio between the characteristic absorption peaks at 2919 cm^−1^ (CH_2_ stretching) and 800 cm^−1^ (Si–O) in the ATR-FTIR spectra was used. It was observed that the normalized ratios (A_2919_/A_800_) increased with the increasing the APTES amount and, for ratios above 5:1, the filler was no longer able to form physical bonds with the reagent having reached saturation. SEM-EDS analysis also shows an increase in nitrogen and carbon content due to the APTES amine groups and propyl chains, respectively, when greater APTES amounts were used. In Figure 6, the EDS spectrum and SEM image of the ARCA scaffold obtained using an APTES:nano-silica ratio of 5:1 was reported. It was possible to notice how RHA nano-silica maintained a spherical morphology during the formation of particle aggregates. Such aggregates caused only a slight decrease in the system porosity as evidenced by liquid displacement measurements (ε ≅ 65%).

The amination of the RHA nano-silica introduced into the scaffold was also confirmed by the comparison of ATR-FTIR spectra of the APTES reagent, RCA, and ARCA scaffolds (Figure 7A,B). An increase in intensity of the bands in the range 2900–2800 cm^−1^ due to antisymmetrical and symmetrical stretching vibrations of the aminopropyl groups of the APTES molecule, a new peak at 1310 cm^−1^, due to wagging and twisting of methylene groups, and a weak band at 690 cm^−1^, attributed to the N–H bending, were evidenced in the spectrum of the scaffold containing aminated RHA nano-silica [56]. 

### 3.4. Immobilization of Laccase on Silica Functionalized Scaffold

According to the experimental results, the immobilized activity increased with the increasing enzyme amount until 0.77 U of laccase per mg of scaffold were used. The immobilization was attempted on four different types of supports: CA, APTES/chitosan/alginate (ACA), RCA, and ARCA scaffolds. The experimental results have shown that the amount of immobilized laccase was highly affected by the APTES presence on the scaffolds. Indeed, the ACA scaffold (immobilized activity = 1.9 U/g) and ARCA scaffold (3.8 U/g) were more active if compared with their precursors without APTES, CA scaffold (0.70 U/g) and RCA (2.63 U/g). Even in terms of immobilization efficiency (%), the ARCA scaffold proved to be the best solid biocatalyst, as shown in Table 2. In Table 2 it is also possible to see how the efficiencies (%) are not very high. Considering that immobilization efficiency represents the fraction of enzyme activity exhibited by the immobilized enzyme relative to the activity removed during the immobilization (see Section 2.10), this parameter is strongly influenced by the possible enzyme inactivation during the immobilization procedure itself. A low efficiency (%) can be explained in two different ways: (i) inactivation of the enzyme in the immobilization mixture, and (ii) deactivation of the enzyme after its immobilization on the support. The first case may be due to the immobilization conditions (temperature, stirring, etc.), while in the second case, the various interactions formed may be the cause as they can modify the protein conformation or lead to protein multilayers which hinders substrate diffusion [57]. However, immobilization continues to be a very advantageous practice thanks to the possibility of reuse and the greater stability of the enzyme, which in any case lead to better performance compared to the soluble biocatalyst.

APTES is a bifunctional reagent, and, consequently, it can interact via hydrogen bonds (siloxane bonds with NH_2_) and electrostatic interactions (–NH_3_^+^ group). It is therefore possible to hypothesize that APTES was incorporated into the RCA scaffold by hydrogen bond-like interactions with the SiOH groups without the involvement of the amino groups, while, considering the CA scaffold, the ionic interactions with the carboxylic groups of alginate with –NH_2_ group of APTES could take place. However, independently of the interaction nature, APTES presence was fundamental in obtaining a high immobilized laccase activity.

In addition, to verify if the size of the filler introduced into the scaffold could influence the activity of the final composites, RHA nano-silica (<0.1 µm, 34 Å pore size) and commercial micro-silica (<5 µm, 100 Å pore size) were used for the preparation of APTES/silica scaffolds. The results showed that the smaller the particles, the higher the immobilized activity (3.8 and 2.7 U/g for the systems containing RHA nano-silica and micro-silica, respectively). The presence of commercial micro-silica in the scaffold structure led to a significant decrease (from 71% to 50%) in the porosity of the composite material, whereas the addition of nano-silica unaffected the native scaffold porosity. This effect, together with the different pore sizes, may be the cause of the difference in activity, as RHA nano-silica results in lower diffusional limitations and maximum surface area per unit of mass, allowing a more effective laccase immobilization.

To evaluate the operational stability of the optimized solid biocatalyst (ARCA scaffold), its reusability was tested for six cycles of ABTS oxidation reaction. Figure 8 shows that the activity of the immobilized enzyme decreased by only 10% at the second use and, despite the more marked decrease in the other cycles, it was possible to carry out another four reuses of the biocatalyst with an activity of 50% compared to the initial one.

Finally, since immobilized laccases have been widely used for the degradation of phenols and resistant dyes [8], the catalytic properties of the ARCA scaffold-based solid biocatalyst were tested using syringic acid as a reducing substrate. The syringic acid conversion was confirmed by UV-Vis analysis of the reaction medium at different times. The results (Figure 9) show that initially (0 min, syringic acid standard solution before reaction with laccase), only a band around 260 nm is visible, which disappears over time in favor of the formation of two new bands at 290 and 360 nm. The initial band is related to syringic acid absorption, while the two new bands can be associated with the quinoid compounds produced by the oxidation reaction such as 2,6-dimethoxy-1,4-benzoquinone [58]. These results lead to believe that the solid biocatalyst completely oxidized the phenol after 24 h (1440 min). Furthermore, the formation of the two new well-defined bands and two isosbestic points at 240 nm and 275 nm observed are in accordance with data reported by Shin [59].

However, since it was difficult to determine the absorbance interference of the product on that of syringic acid, the conversion % was determined more precisely by HPLC-DAD. The obtained results showed the absence of the syringic acid chromatographic peak (4.55 min) after 24 h and the presence of a new peak (7.93 min) (100% conversion after 24 h) probably due to the quinoid oxidation products [60]. In addition, since no change in the UV spectra of the control (ARCA scaffold alone) was evidenced, it was possible to confirm that the total degradation pattern was only related to the enzymatic action.

## 4. Conclusions

For the first time, a cross-linked chitosan/alginate scaffold containing RHA nano-silica was successfully used for laccase immobilization to obtain an efficient, cheap, and easy-to-use solid biocatalyst. Amorphous nano-silica (99.6% purity and 70–100 nm particle size) was obtained from the agro-industrial waste rice-husk ash (RHA) employing a thermal treatment at 550 °C and alkaline extraction. The introduction of RHA nano-silica in the CA scaffold significantly increased its thermal and mechanical properties without altering its porosity. To increase laccase loading, an amination reaction with APTES of the RHA nano-silica contained in the scaffold was carried out. The enzyme immobilization in a conformation favorable for the interaction with the substrate was evidenced by the good activity of the system (3.8 U/g). Despite the low immobilization efficiency (5.3%), the excellent performance of the developed biocatalyst was demonstrated by both its reusability (50% up to the 6th cycle) and the syringic acid’s complete oxidation in 24 h.

In conclusion, our results demonstrated how a composite system based on polysaccharides and rice-husk-derived nano-silica can be an optimal support for a laccase-based solid biocatalyst to be used for bioremediation. This new green approach could also make it possible to recycle a waste that represents an environmental hazard due to its uncontrolled disposal using it in the synthesis of a material with enormous added value such as nano-silica.

## Figures and Tables

**Figure 1 polymers-15-03127-f001:**
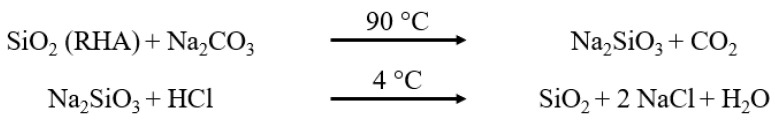
Chemical reaction of the nano-silica extraction process from RHA.

**Figure 2 polymers-15-03127-f002:**
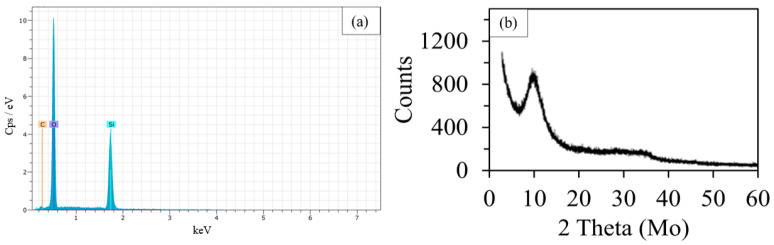
EDS analysis (**a**) and X-ray diffraction graph (**b**) of rice-husk nano-silica. The X-ray spectrum was obtained using Mo as a target material.

**Figure 3 polymers-15-03127-f003:**
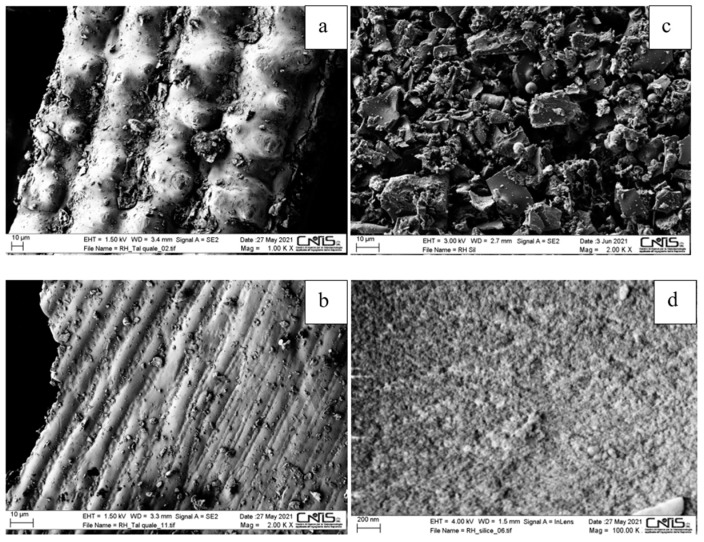
SEM morphological analysis of rice husk (**a**,**b**) and rice-husk nano-silica (**c**,**d**) at different magnification (1, 2, 2, and 100 KX, respectively).

**Figure 4 polymers-15-03127-f004:**
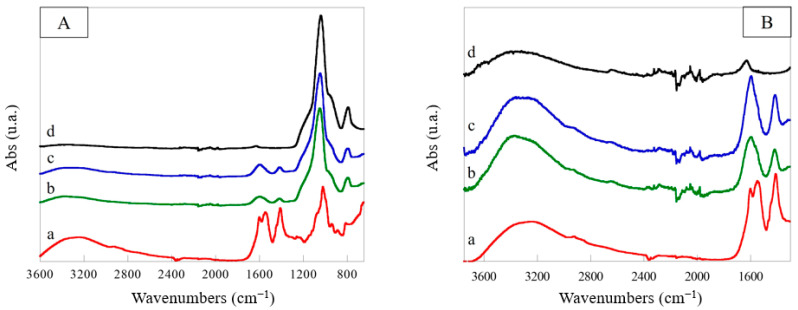
ATR-FTIR spectra (**A**) of chitosan/alginate scaffold (CA) (a), RHA nano-silica/chitosan/alginate (RCA) scaffolds with nano-silica:scaffold ratio of 1:1 (w:w) (b) and 2:1 (w:w) (c), and RHA nano-silica (d) and their magnification (**B**).

**Figure 5 polymers-15-03127-f005:**
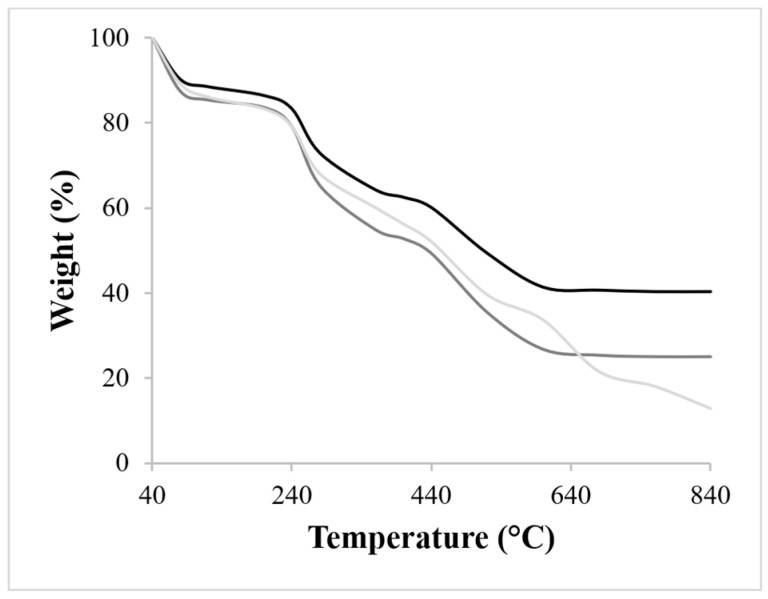
Thermogravimetric curves of CA (grey line), RCA 2:1 (w:w) (black line), and RCA 1:1 (w:w) (dark grey line) scaffolds.

**Figure 6 polymers-15-03127-f006:**
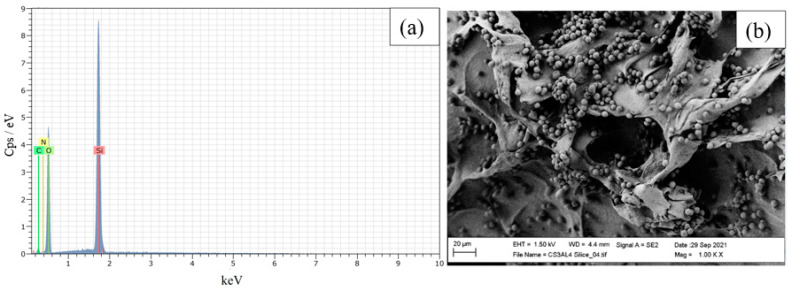
EDS spectra (**a**) and SEM micrograph (**b**) of APTES/RHA nano-silica/chitosan/alginate scaffold obtained using an APTES:nano-silica ratio of 5:1 (w:w).

**Figure 7 polymers-15-03127-f007:**
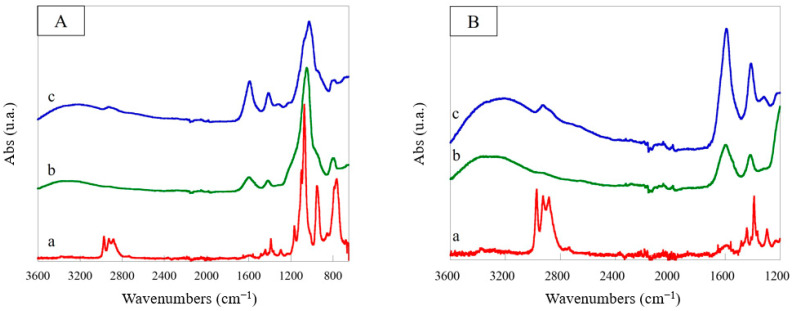
ATR-FTIR spectra (**A**) of APTES (a), RCA scaffold (b), ARCA scaffold (APTES:nano-silica:scaffold ratio of 10:2:1 (w:w:w) (c), and their magnification (**B**).

**Figure 8 polymers-15-03127-f008:**
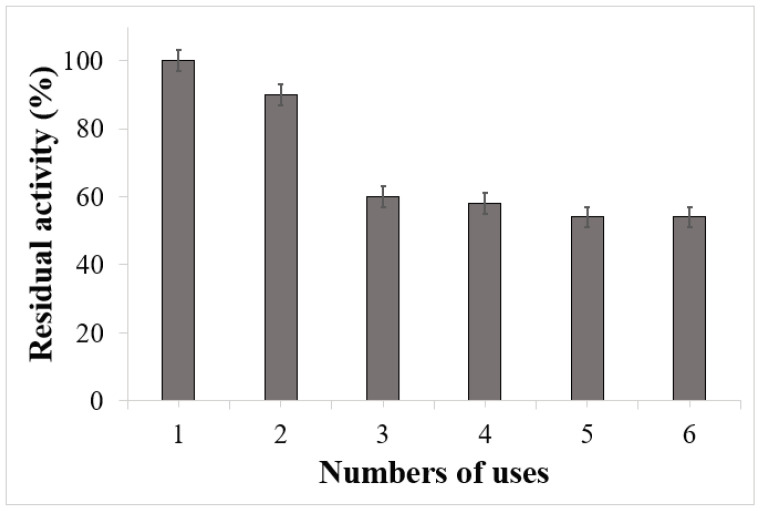
Operational stability of the laccase-APTES/RHA nano-silica/chitosan/alginate scaffold. Experimental conditions: 0.18 mM ABTS, 30 °C, 0.1 M citrate—0.2 M PBS, pH 3.

**Figure 9 polymers-15-03127-f009:**
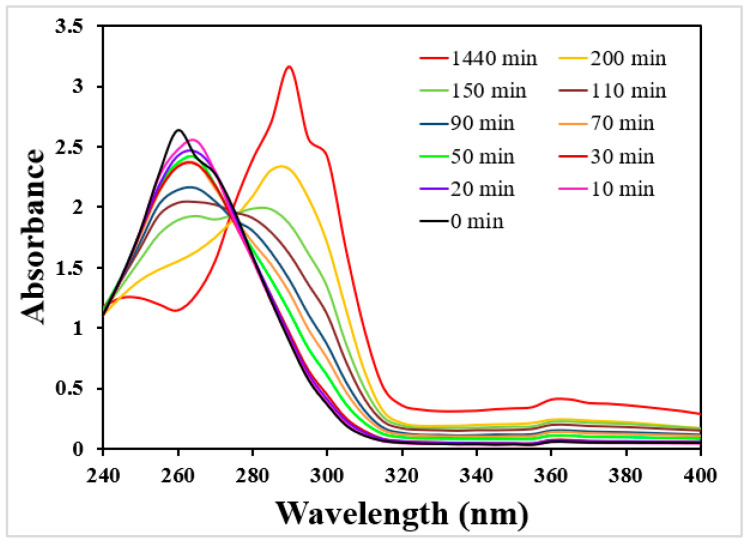
UV-Vis spectra of the syringic acid in the reaction medium at different times of incubation with the ARCA scaffold-optimized biocatalyst. Experimental conditions: starting biocatalyst activity 0.013 U, analysis time 24 h (1440 min), syringic acid concentration 50 mg/L, pH 5.

**Table 1 polymers-15-03127-t001:** Elemental analysis of rice husk (RH) and rice-husk ash (RHA) at different calcination times (T = 550 °C).

Element (%)	0 h	2 h	3 h	4 h	5 h
C	38.70	3.80	1.18	0.40	0.04
H	6.00	0.44	0.49	0.29	0.38
N	0.45	0.37	0.11	0	0

**Table 2 polymers-15-03127-t002:** Immobilization parameters of the solid biocatalysts synthesized in the optimal immobilization conditions: pH 7, immobilization time = 1.5 h, room temperature and 0.77 U/mg of support.

Scaffold	Immobilized Activity (U/g)	Efficiency (%)
Chitosan/Alginate (CA)	0.7	0.9
APTES/Chitosan/Alginate (ACA)	1.9	1.9
RHA/Chitosan/Alginate (RCA)	2.6	3.1
APTES/RHA/Chitosan/Alginate (ARCA)	3.8	5.3

## Data Availability

The authors confirm that the data supporting the findings of this study are available within the article.

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
