# Peer review of "Design of a 3D Amino-Functionalized Rice Husk Ash Nano-Silica/Chitosan/Alginate Composite as Support for Laccase Immobilization"

_polymers, 2023, doi:10.3390/polym15143127_

Round 1

Reviewer 1 Report

The manuscript describes the design and synthesis of a 3D amino-functionalized rice husk ash composite as a support for laccase immobilization. Characterization techniques, including FTIR, XRD, SEM, and EDS, were utilized. The authors have conducted extensive work for this project. However, there are still some issues that need to be addressed:

  1. What are the novelties of this study? As the authors emphasized in the manuscript, it was the first time rice husk ash was used as a source of nano-silica. However, both the synthesis of nano-silica and the composite are not new to the community. I fail to see the importance and necessity of using rice husk as the source of nano-silica.
  2. Building upon the previous point, the authors have conducted a significant amount of work in this study. However, most of the discussions only involve the basic characterization of the synthesized materials. The data does not appear to be surprising or novel to the readers. Furthermore, similar studies have been reported by others previously, and I cannot identify any unique aspects compared to the referenced literature.
  3. I would suggest that the authors spend more time identifying the novel aspects of their study. The current manuscript does not seem particularly interesting to readers.
  4. The background section contains a large amount of information covering several different fields. It would be advisable to rewrite the background section, emphasizing the most important points.

Based on the current version of the manuscript, I recommend rejecting it.

Author Response

Reviewer 1

We would like to thanks the Reviewer for the interesting and helpful comments which allowed to improve the quality of the manuscript.

According to reviewer’s suggestions, the manuscript was carefully revised.

The manuscript describes the design and synthesis of a 3D amino-functionalized rice husk ash composite as a support for laccase immobilization. Characterization techniques, including FTIR, XRD, SEM, and EDS, were utilized. The authors have conducted extensive work for this project. However, there are still some issues that need to be addressed:

  1. What are the novelties of this study? As the authors emphasized in the manuscript, it was the first time that rice husk ash was used as a source of nano-silica. However, both the synthesis of nano-silica and the composite are not new to the community. I fail to see the importance and necessity of using rice husk as the source of nano-silica.

Although, as the referee rightly pointed out, both the synthesis of nano-silica and polymeric composites are nothing new, this is the first time that a polyelectrolyte complex based on polysaccharides has been combined with nano-silica obtained from agro-industrial waste by an inexpensive (500°C) process to produce a carrier for laccase immobilization for bioremediation. The production of nano-silica from rice husks is an enormous issue as common methods of silica extraction are thought to be dangerous to the environment due to intensive use of chemicals, energy and non-renewable resources. For these reasons, the scientific research is currently focused on the use of new feedstocks for a greener silica production. Therefore, our approach could make it possible to recycle an agro-industrial waste that represents an environmental hazard due to its uncontrolled burning.

In the revised manuscript these aspects were introduced in the introduction section (please, see manuscript).

  1. Building upon the previous point, the authors have conducted a significant amount of work in this study. However, most of the discussions only involve the basic characterization of the synthesized materials. The data does not appear to be surprising or novel to the readers. Furthermore, similar studies have been reported by others previously, and I cannot identify any unique aspects compared to the referenced literature.

We agree to the Reviewer. In literature, there are other studies on the characterization of nano-silica. However, in our case, given the need to optimize the production process of nano-silica from rice husk ash, particularly purification and calcination phases which are different from those existing in the literature, it was necessary to carry out an important characterization work of the product obtained for the successful immobilization of laccase.

  1. I would suggest that the authors spend more time identifying the novel aspects of their study. The current manuscript does not seem particularly interesting to readers.

The introduction was totally rewritten. We hope that the novel aspects of the work have been highlighted in the revised manuscript (please, see manuscript).

  1. The background section contains a large amount of information covering several different fields. It would be advisable to rewrite the background section, emphasizing the most important points.

The background section has been extensively rewritten to more clearly present and emphasize the most important points (please, see manuscript).

Reviewer 2 Report

The article « Design of a 3D amino-functionalized rice husk ash nano-silica/chitosan/alginate composite as support for laccase immobilization» is attracted to a relevant topic and has a high applied value.

The article is well structured, written in sufficient detail and logically.

Comments

1. The authors synthesized CA scaffold by forming of chitosan-alginate IPEC with following addition of calcium salt. How the authors confirm, that after Ca2+ addition total IPEC destruction doesn't occur? May be final product is 3D alginate gelated by calcium with interpenetrated chitosan chains?
2. 1692 cm-1 is not amide I for chitosan, is a complex band reflecting vibrations of different bonds. Also, 1590 cm-1 is not position of shifted amide II, it is stretching of COO-. Revise FTIR description.
3. Line 372: why the authors conclude that SiOH interact with CA scaffold through C=O groups? Content of the latter is extremely less compared to OH groups of polysaccharides.
4. Line 403: "observe new band at 2800-2900". These bands are in the spectrum of CA scaffold due to vibrations of CH2 groups of pyranose rings. Revise this part

5. Table 2 shows the very small immobilization efficiency values (%). The authors should explain why this happened.

 Minor editing of English language required

Author Response

Reviewer 2

We would like to thank the Reviewer for the interesting and helpful comments which allowed to improve the quality of the manuscript.

According to reviewer’s suggestions, the manuscript was carefully revised.

The article « Design of a 3D amino-functionalized rice husk ash nano-silica/chitosan/alginate composite as support for laccase immobilization» is attracted to a relevant topic and has a high applied value. The article is well structured, written in sufficient detail and logically.

  1. The authors synthesized CA scaffold by forming of chitosan-alginate IPEC with following addition of calcium salt. How the authors confirm, that after Ca2+ addition total IPEC destruction doesn't occur? May be final product is 3D alginate gelated by calcium with interpenetrated chitosan chains?

CA scaffolds, at any molar ratio of two components, are unstable in an aqueous medium and flake easily. The crosslinking reaction with CaCl2 is necessary to improve their dimensional stability and mechanical properties. This reaction can be carried out on the coacervate in suspension or on the scaffold. We verified that if the crosslinking with CaCl2 was carried out after formation of the three-dimensional structure by freeze-drying, the scaffolds were stable for several months. As reported in the literature (Wang l, Khor E, Lim LY. J. Pharm. Sci., 90, 1134-1142 (2001)), a possible displacement of the chitosan-alginate bond from CaCl2 can occur easily in the coacervate in suspension rather than in its solid form, where the conformational rotations and relaxations of the polymer chains are hampered. In addition, in our work, to facilitate the formation of PECs, the scaffolds were fabricated using a greater amount of alginate than chitosan (chitosan:alginate 1:1.7 molar ratio). Therefore, we can reasonably think that a solid structure containing several chitosan-alginate interactions subsequently stabilized by ionic interactions between CaCl2 and -COOH excess groups of alginate was formed.

  1. 1692 cm-1 is not amide I for chitosan, is a complex band reflecting vibration of different bonds. Also, 1590 cm-1 is not position of shifted amide II, it is stretching of COO-. Revise FTIR description.

We agree with the Reviewer. The assignments of the two complex bands in the range 1700-1500 cm-1 are incorrect. However, we have noticed that the IR spectra reported in figure 4 are also- incorrect. Particularly, a scale shift of about 100 cm-1 is present in all spectra. Therefore, the two bands for which the attributions have been revised are present at about 1604 and 1545 cm-1. Such peaks were respectively assigned to the carboxylic groups of the alginate which interact with chitosan (COO- antisymmetric stretching, antisymmetric bending of the protonated amine and amide I at 1604 cm-1) and to the amino groups of chitosan which interact with alginate (N-H amine bending, amide II and symmetric -NH3+ bending at 1545 cm-1).

Therefore, the text in the revised manuscript (lines 359-379) was changed with:

“In Figure 4A, the ATR-FTIR spectra of the scaffolds containing RHA nano-silica at two nano-silica:scaffold ratios, 1:1 and 2:1 (w:w), in comparison with those of CA and pristine nano-silica were reported. In the spectrum of the CA scaffold can be highlighted an intense band in the range 3600-3000 cm-1 attributed to the stretching of the -OH groups of the two polysaccharides, and two peaks at 1604 and 1545 cm-1 assigned to the carboxylic groups of the alginate which interact with chitosan (-COO- antisymmetric stretching and antisymmetric bending of the protonated amine and amide I, at 1604 cm-1) and to the amino groups of chitosan which interact with alginate (N-H amine bending, amide II and symmetric -NH3+ bending, at 1545 cm-1). The carboxylic groups of the alginate also showed a symmetric stretching at 1410 cm-1. Finally, in the range 1180-800 cm-1, the C-O-C, C-O, and C-O-H stretching bands due to the pyranose rings were evidenced. After introduction of RHA nano-silica, some changes were observed in the spectra of the modified scaffold. First of all, the peaks attributed to the antisymmetric and symmetric stretching of the Si-O-Si group were evident at 1040 and 800 cm-1, respectively. Furthermore, the remarkable decrease of the intensity of the bands at 1410 cm-1 and 1545 cm-1, with the peak at 1545 cm-1 becoming a shoulder of the band at 1604 cm-1 (see magnification shown in figure 4B), confirmed the hydrogen bonding interactions between the SiOH groups and carbonyl and amino groups of the CA scaffold. In particular, the decrease of the band intensity at 1545 cm-1 results correlated to the nano-silica content in the structure (scaffold with a nano-silica:scaffold ratio of 2:1) demonstrating an increase in the number of hydrogen bonds established between the SiOH and amino groups of chitosan with an increase of the filler.”

Line 372: why the authors conclude that SiOH interact with CA scaffold through C=O groups? Content of the latter is extremely less compared to OH groups of polysaccharides.

The sentence (lines 370-372) was changed in: “By comparing TGA thermograms, it was possible to observe that RHA nano-silica is able to make the scaffolds more thermally stable (Figure 5), probably due to interactions between its –OH groups and the ones of the polymers (-OH, -NH and -CO).”

  1. Line 403: "observe new band at 2800-2900". These bands are in the spectrum of CA scaffold due to vibrations of CH2 groups of pyranose rings. Revise this part.

We agree with the Reviewer’s comment. Again, Figure 7 has an error. In fact, the spectrum b shown in the figure is not that of RCA scaffold (CA scaffold containing nano-silica) but that of the CA scaffold pristine (see figure 4). In the revised manuscript, we inserted in the figure 7 the correct spectrum (figure 7A) and also a magnification (figure 7B) of the spectra in the range 4000-1200 cm-1. In the spectrum of the CA scaffold containing nano-silica (spectrum c), the contribution of APTES is now more evident with the increase in intensity of the bands in the range 2900-2800 cm-1 and the formation of a new band due to wagging and twisting of methylene groups at 1310 cm-1.

Therefore, the text in the manuscript (lines 417-421 and lines 439-443) was changed in:

(lines 417-421): “The RHA nano-silica amination was confirmed by ATR-FTIR spectra, where it was possible to observe an increase in intensity of the bands at 2919 cm−1 and 2800 cm−1 and the formation of a new peaks at 1310 cm-1 (due to contribution of CH2 stretching and bending of propyl chain in APTES, respectively) [50]”.

(lines 439-443): “An increase in intensity of the bands, in the range 2900-2800 cm-1, due to antisymmetrical and symmetrical stretching vibrations of aminopropyl groups of APTES molecule, a new peak at 1310 cm-1, due to wagging and twisting of methylene groups, and a weak band at 690 cm-1, attributed to the N-H bending, were evidenced in the spectrum of the scaffold containing aminated RHA nano-silica [56].”

  1. Table 2 shows the very small immobilization efficiency values (%). The authors should explain why this happened.

An explanation for the low immobilization efficiency values was inserted.

“In Table 2 it is also possible to see how the efficiencies (%) are not very high. Considering that immobilization efficiency represents the fraction of enzyme activity exhibited by the immobilized enzyme relative to the activity removed during the immobilization (see section 2.10), this parameter is strongly influenced by the possible enzyme inactivation during the immobilization procedure itself. A low efficiency (%) can be explained in two different ways: i) inactivation of the enzyme directly in the immobilization mixture, and ii) deactivation of the enzyme after its immobilization on the support. The first case may be due to the immobilization conditions (temperature, stirring, etc.), while in the second case the various interactions formed can be the cause as they can modify the protein conformation or lead to protein multilayers which hinders substrate diffusion [57]. However, immobilization continues to be a very advantageous practice thanks to the possibility of reuse and the greater stability of the enzyme which in any case lead to better performance compared to the soluble biocatalyst.”

Reviewer 3 Report

The paper is devoted for 3D amino-functionalized rice husk ash nanosilica/chitosan/alginate composite preparation and characterization for laccase immobilization. The topic is generally interesting, however the paper contain unexplained places (below) and need major revisions.

1 1) The aim of the paper should be more clearly formulated.

2 2) Graphical quality of Figs. 2 and 4 should be improved.

3 3)  Fig. 4, why Infrared investigations were performed only in frequency range 4000-500 cm-1? The absoption band close to 3500 cm-1 for lines a, b and c is not commented in the paper text.

44) Fig. 7 please comment the absorption band close to 1000 cm-1.

55) Fig. 9 should be more discussed. It should be color online.

66)  Conclusions should be rewritten in more informative way.

Author Response

Reviewer 3

We would like to thanks the Reviewer for the interesting and helpful comments which allowed to improve the quality of the manuscript.

According to reviewer’s suggestions, the manuscript was carefully revised.

  1. The aim of the paper should be more clearly formulated.

As suggested from the Reviewer, the aim of the work was better described. The introduction was totally revised (please, see revised manuscript).

  1. Graphical quality of Figs. 2 and 4 should be improved.

The graphical quality of figures 2 and 4 was improved.

  1. 4, why Infrared investigations were performed only in frequency range 4000-500 cm-1? The absorption band close to 3500 cm-1 for lines a, b and c is not commented in the paper text.

Spectra in ATR mode were recorded in the range 4000-650 cm-1 because ZnSe lenses that focus the IR beam on the internal reflection element absorb below 650 cm-1.

  1. 7 please comment the absorption band close to 1000 cm-1.

The discussion regarding FTIR analysis was totally revised and some spectra reported in the figures 4 and 7 changed. Therefore, the band close to 1000 cm-1 was commented. Following the new text inserted in the revised manuscript (lines 359-379):

“In Figure 4A, the ATR-FTIR spectra of the scaffolds containing RHA nano-silica at two nano-silica:scaffold ratios, 1:1 and 2:1 (w:w), in comparison with those of CA and pristine nano-silica were reported. In the spectrum of the CA scaffold can be highlighted an intense band in the range 3600-3000 cm-1 attributed to the stretching of the -OH groups of the two polysaccharides, and two peaks at 1604 and 1545 cm-1 assigned to the carboxylic groups of the alginate which interact with chitosan (-COO- antisymmetric stretching and antisymmetric bending of the protonated amine and amide I, at 1604 cm-1) and to the amino groups of chitosan which interact with alginate (N-H amine bending, amide II and symmetric -NH3+ bending, at 1545 cm-1). The carboxylic groups of the alginate also showed a symmetric stretching at 1410 cm-1. Finally, in the range 1180-800 cm-1, the C-O-C, C-O, and C-O-H stretching bands due to the pyranose rings were evidenced. After introduction of RHA nano-silica, some changes were observed in the spectra of the modified scaffold. First of all, the peaks attributed to the antisymmetric and symmetric stretching of the Si-O-Si group were evident at 1040 and 800 cm-1, respectively. Furthermore, the remarkable decrease of the intensity of the bands at 1410 cm-1 and 1545 cm-1, with the peak at 1545 cm-1 becoming a shoulder of the band at 1604 cm-1 (see magnification shown in figure 4B), confirmed the hydrogen bonding interactions between the SiOH groups and carbonyl and amino groups of the CA scaffold. In particular, the decrease of the band intensity at 1545 cm-1 results correlated to the nano-silica content in the structure (scaffold with a nano-silica:scaffold ratio of 2:1) demonstrating an increase in the number of hydrogen bonds established between the SiOH and amino groups of chitosan with an increase of the filler.”

  1. 9 should be more discussed. It should be color online.

The figure 9 was better discussed and the curves were reported in color.

The text introduced was:

“The results (Figure 9) show that initially (0 min, syringic acid standard solution before re-action with laccase) is visible only a band around 260 nm which disappears over time in favor of the formation of two new bands at 290 and 360 nm. The initial band is related to syringic acid absorption, while the two new bands can be associated to the quinoid compounds produced by the oxidation reaction such as 2,6-dimethoxy-1,4-benzoquinone [58]. These results lead to believe that the solid biocatalyst completely oxidized the phenol after 24 h (1440 min). Furthermore, the formation of the two new well-defined bands and two isosbestic points at 240 nm and 275 nm observed are in accordance with data reported by Shin [59].

  1. Conclusions should be rewritten in more informative way.

As suggested by the Reviewer, the conclusions were rewritten in more informative way.

Round 2

Reviewer 1 Report

Thanks for the revisions.

The further comments along with the updated manuscript address most of my questions. I do have some one minor suggestion regarding the updated manuscript:

1. The EDS spectra seems blurry and the formats are not consistent. Please make them better if possible.

Author Response

Review #1

Thanks for the revisions.

The further comments along with the updated manuscript address most of my questions. I do have one minor suggestion regarding the updated manuscript: We would like to thank the reviewer for his work and express our pleasure to know that he found our answers and revision satisfactory.

  1. The EDS spectra seem blurry and the formats are not consistent. Please make them better if possible.

The images portraying the EDS spectra have been improved in quality and we made sure that each figure had the same font for the axis titles.

Reviewer 3 Report

Authors make proper corrections according to reviewer remarks

and I suggest to publish the paper as it is.

Author Response

Review #3

Authors make proper corrections according to reviewer remarks and I suggest to publish the paper as it is.

We warmly thank this reviewer for his statement and previous advice.